# Nasal Symptoms in Asthmatic Patients under Treatment with Anti-IL-5 Monoclonal Antibodies. A Real-Life Cohort Study

**DOI:** 10.3390/jcm11237056

**Published:** 2022-11-29

**Authors:** Juan Maza-Solano, Christian Calvo-Henríquez, Isam Alobid, Marta Álvarez-Cendrero, Óscar Palomares, Ramón Moreno-Luna, Jaime Santos-Perez, Jaime González-García, Serafín Sánchez-Gómez

**Affiliations:** 1Rhinology Study Group, Young-Otolaryngologists of the International Federations of Oto-Rhino-Laryngological Societies (YO-IFOS), 13005 Paris, France; 2Rhinology Unit, Department of Otolaryngology, Head and Neck Surgery, Virgen Macarena University Hospital, 41003 Sevilla, Spain; 3Service of Otolaryngology, Rhinology Unit, Hospital Complex of Santiago de Compostela, 15706 Santiago de Compostela, Spain; 4Rhinology and Skull Base Unit, ENT Department, Hospital Clínic, Instituto de Investigaciones Biomédicas August Pi i Sunyer (IDIBAPS), CIPERES, Barcelona University, 08007 Barcelona, Spain; 5Department of Biochemistry and Molecular Biology, School of Chemistry, Complutense University of Madrid, 28040 Madrid, Spain; 6Service of Otolaryngology, Rhinology Unit, University Hospital of Valladolid, 47005 Valladolid, Spain

**Keywords:** asthma, benralizumab, chronic rhinosinusitis, mepolizumab, nasal polyps, reslizumab

## Abstract

Currently, some monoclonal antibodies (mAbs) are being studied for chronic rhinosinusitis with nasal polyps (CRSwNP). Three anti-IL-5 mAb: mepolizumab, reslizumab and benralizumab, have been tested through randomized clinical trials. In this real-life study, we aimed to describe the nasal effects of a cohort of asthmatic adults treated with anti-IL-5 mAb. Methods: We carried out an observational study in adults (≥18 years) on anti-IL-5 mAb treatment. Variables included ACT and SNOT−22 questionnaires, nasal polyps score, blood total IgE levels and blood eosinophil count. Results: Overall, 38 participants were included in the study; 19 patients received mepolizumab, 17 were treated with benralizumab and 2 patients were given reslizumab. There was a statistically significant difference in the ACT and SNOT−22 scores before and after mAb treatment. ACT score increased from 11.05 to 21.5 after treatment (*p* < 0.001). SNOT−22 decreased from 57 to 37.3 after treatment (*p* = 0.004). No statistically significant differences between mAb groups were observed regarding the ACT or the SNOT−22 (*p* = 0.775) response (*p* = 0.775). In addition, 60.53% of patients obtained a minimal clinically important difference (MCID) in SNOT−22. Conclusions: A significant clinical response based on SNOT−22 score evolution after anti-IL-5 mAb treatment was observed. This study also demonstrated that blood eosinophil count, rather than serum total IgE levels, is the best predictor of asthma symptom improvement, which was assessed through the ACT and SNOT−22 questionnaires.

## 1. Introduction

Chronic rhinosinusitis with nasal polyps (CRSwNP) is a chronic inflammation of the nasal epithelia and paranasal sinus. According to recent epidemiological studies, CRSwNP may affect up to 12% of the adult population [1] CRSwNP exacerbates lower airway disease, leading to a poorer quality of life and increased direct and indirect medical costs [1]. CRSwNP is mainly managed by topical and/or oral corticosteroids. Traditionally, endoscopic sinus surgery (ESS) is considered in CRSwNP patients who fail to improve after medical treatment [2]. In recent years, a new CRSwNP treatment line has been introduced, the monoclonal antibodies (mAb), that target mediators of the type 2 inflammation route [3]. The interest in mAb therapy for CRSwNP arose from the experience in asthmatic patients. As asthma is closely linked to CRSwNP, it was shortly evidenced that mAb therapy was also efficient in reducing nasal polyps and symptoms.

There are two main routes to Type 2 inflammation; the first involves eosinophils and IL-5, while the second encompasses IL-4 and IL-13-induced immunoglobulin E (IgE). In white patients, approximately 80% of the nasal polyps are characterized by prominent eosinophilia. Eosinophils may contribute to tissue damage and polyp growth by releasing toxic products [4]. Different mAbs targeting specific molecules involved in type 2-mediated inflammatory pathways have been approved to treat CRSwNP. Among these mAbs, dupilumab6 (anti-IL-4Rα mAb inhibiting both IL-4 and IL-13), omalizumab7 (anti-IgE mAb), and mepolizumab8 (anti-IL-5 mAb) have been licensed by the Food and Drug Administration (FDA) and/or European Medicines Agency (EMA) for CRSwNP control. Two other anti-IL-5 mAb, reslizumab9 (anti-IL-5 mAb) and benralizumab [5] (anti-IL-5Rα mAb), were tested through randomized clinical trials but have not yet been approved by the FDA or EMA.

In this real-life study, we aimed to describe the nasal effects of a cohort of asthmatic adults treated with anti-IL-5 mAb for CRSwNP.

## 2. Materials and Methods


**
Sample
**


An observational study was undertaken between September 2017 and March 2021.

We recruited adults (≥18 years) on anti-IL-5 mAb treatment for airway inflammatory disease from the asthma unit of a third referral university hospital (Seville, Spain). The asthma unit consists of pneumologists, allergists and otolaryngologists. Eligible participants were severe asthmatic patients receiving nasal and inhaled corticosteroid therapy.

We excluded from the study patients with any of the following characteristics: severe uncontrolled asthma under biological treatment other than anti-IL-5, individuals under 18 years of age, patients undergoing cancer treatment, patients with Churg–Strauss syndrome, and/or pregnant or lactating women. Patients without CRS but receiving anti-IL-5 were still included in this study.

We collected the following clinical data: history of ESS, adenoidectomy, and/or tonsillectomy, blood pressure (high or under treatment), cholesterol level (hypercholesterolemia or under treatment), weight before and after mAb treatment, height, body mass index (BMI), sex, having any of the following diseases: asthma, sleep apnea, emphysema, bronchiectasis, heart disease, chronic obstructive pulmonary disease (COPD), hypertoroids and/or gastroesophageal reflux, upper respiratory tract infections (tonsillitis, adenoiditis, pharyngitis, otitis), blood total IgE level and blood eosinophils (determined automatically using a 2-mL heparinized blood sample). We also inquired about smoking habits and alcohol consumption.


**
Physical Examination
**


The physicians JMS, RML, and JGG examined the patients who attended the outpatient clinics of the Rhinology Department with suggestive symptoms of sinonasal pathology (nasal congestion, rhinorrhea, smell alterations, facial pain, sneezing, etc.). All participants underwent nasal flexible endoscopy (Olympus Flexible: Visera-Elite-190). The following clinical characteristics were recorded: nasal mucosa color (red, pale, or normal), inferior and middle turbinate size and nasal polyp score (NPS). Polyps were scored according to Meltzer scoring system.11 Meltzer scale ranges from 0 to 4: 0 being no polyps, 1 being polyps confined to the middle meatus, 2 being multiple polyps occupying the middle meatus, 3 being polyps extending beyond the middle meatus, and 4 being polyps completely obstructing the nasal cavity.


**
Patient Symptoms
**


All participants were requested to complete these two questionnaires: the asthma control test (ACT) and the sinonasal outcome test (SNOT−22). At the Rhinology Department, JMS, RML and JGG provided patients with the Spanish version of the questionnaires. ACT is a self-administered tool for identifying patients with poorly controlled asthma, while SNOT−22 assesses different nasal symptoms. ACT and SNOT−22 consist of 5 and 22 items, respectively, that are answered using a five-point Likert scale. Both questionnaires were completed before and after mAb treatment.

The minimal clinically relevant difference (MCID) in SNOT−22 was defined to be 12 points, according to recently published normative data [6].


**
mAb Therapy
**


The biological therapy was chosen according to the Spanish Guideline on the Management of Asthma (GEMA).14 This guideline specifies that severe uncontrolled asthma patients with a blood eosinophil level >300 cells/μL in the past 12 months or with a current blood eosinophil count >150 cells/μL are candidates for treatment with mepolizumab, reslizumab or benralizumab. The GEMA guideline does not require any specific clinic criteria to prescribe biologics, only frequent exacerbations and poorly controlled asthma despite appropriate medical treatment (step 5). Physicians decide to prescribe any of those three treatments by referring to the criteria established by the Severe Asthma Commission.

The evaluation of the therapeutic response to mAb is determined at four months from treatment initiation.


**
Ethics Statement
**


The study was performed in accordance with the ethical standards laid down in the Declaration of Helsinki and all patients signed a written informed consent form.


**
Statistical Analysis
**


All quantitative variables were tested for normality with the Shapiro–Wilk test. Comparison between quantitative and dichotomic variables was performed with *t*-test if a normal distribution was demonstrated, otherwise the non-parametric variation Rank Sum test was applied. The relationship between the qualitative variables was studied through a chi-square test. The correlation between the quantitative variables was performed through the Spearman correlation analysis.

## 3. Results


**
Participants
**


A total of 38 participants were included in the study; 19 patients (50%) received mepolizumab, 17 (44.74%) were treated with benralizumab and 2 (5.26%) patients were given reslizumab.

The general characteristics of the study population are summarized in Table 1. There were 22 females (57.89%), and the mean age of the participants was 56.66 ± 9.00 (min/max range: 33–79 year).

Overall, there were no statistically significant differences between patients treated with benralizumab, mepolizumab or reslizumab concerning age, gender, weight, BMI and previous nasal surgery (ESS, adenoidectomy, turbinate surgery or septoplasty). The patients were not comparable regarding inflammatory markers (Table 1). Patients under benralizumab treatment have the highest levels of IgE. They also showed the highest eosinophil counts, yet the difference from the other mAb groups was not statistically significant.

The distribution of the risk factors and comorbid conditions is summarized in Appendix A. There were no statistically significant differences between mAb treatment groups regarding emphysema, bronchiectasis, sleep apnea, asthma, gastroesophageal reflux, diabetes, hypercholesterolemia, hypertension, alcohol consumption, smoking habits and non-steroidal anti-inflammatory drug (NSAID)-exacerbated respiratory disease (N-ERD).


**
Response to Treatment
**


Figure 1 summarizes the data on patients’ response to mAb treatment (N = 38). There were no episodes of adenoiditis, tonsillitis or pharyngitis. Four (10.53%) episodes of otitis were registered. Though otitis was more frequent among patients treated with mepolizumab (15.79%) than among those who received benralizumab (5.88%) or reslizumab (0%), this difference was not statistically significant (*p* = 0.58).

Overall, there was a statistically significant difference in the ACT and SNOT−22 scores before and after mAb treatment. ACT score increased from 11.05 to 21.5 after treatment (*p* < 0.001). SNOT−22 decreased from 57 to 37.3 after treatment (*p* = 0.004). No statistically significant differences between mAb groups were observed regarding the ACT response (*p* = 0.15) or the SNOT−22 (*p* = 0.78) (Figure 1).

Precisely 60.53% of the patients obtained a minimal clinically important difference (MCID) in SNOT−22. There is a difference between groups of mAb therapy, but it is not statistically significant (*p* = 0.18) (Figure 1).

During the follow-up period, there were no episodes of tonsillitis, adenoiditis or pharyngitis. There were four cases of otitis, without statistically significant differences between drugs (chi^2^ 1.08, *p* = 0.584).


**
Predictors of Clinical Response
**


Results are summarized in Table 2. The pretreatment levels of eosinophils positively correlated with the change in the ACT (rho = −0.32; *p* = 0.05) and the SNOT−22 (rho = 0.30; *p* = 0.07) scores, yet the correlation was not statistically significant.

No statistically significant association was observed between the pretreatment levels of serum total IgE and the changes in the ACT (*p* = 0.25) and SNOT−22 (*p* = 0.75) scores before and after mAb treatment.

As for other indicators of response to mAb treatment, the weight before treatment was significantly related to the change in the ACT (rho = 0.29, *p* = 0.103) and SNOT−22 scores (rho = −0.36; *p* = 0.04) (Figure 2). However, correlation with BMI was neither found with ACT (rho = 0.18, *p* = 0.30) nor with SNOT−22 (rho = −0.23, *p* = 0.21) scores.

Other factors were explored to predict the clinical response to anti-IL-5 mAb (N-ERD, previous surgery, sex, smoking habit, comorbid conditions: emphysema, bronchiectasis, sleep apnea, asthma, gastroesophageal reflux, diabetes, hypercholesterolemia and/or hypertension). No statistically significant association was observed between N-ERD and the change in SNOT−22 score before and after mAb treatment. There was a difference between the cohorts with and without N-ERD regarding ACT change, despite not being statistically significant (*p* = 0.07).


**
Nasal Polyp Score (NPS)
**


Despite not being statistically significant, we observed a relationship between NPS and the SNOT−22 change (chi^2^ = 4.69, *p* = 0.20) before and after mAb therapy. Nasal symptoms did not change between before and after mAb therapy in patients with polyp score III (13.11); however, there was a notable improvement in nasal symptoms for patients with polyps scores I (28) and II (32.45).

Table 3 summarizes the relationship between NPS and different clinical features. NPS is related to eosinophils (chi^2^ = 8.86, *p* = 0.03), but not to serum total IgE (*p* = 0.38). There is also a statistically significant association between polyp size and patient´s weight (*p* = 0.004) and BMI (chi^2^ = 7.29, *p* = 0.03). There is no association between polyp size and the other variables (Table 3), including SNOT−22, ACT, or N-ERD.

A subgroup analysis regarding NPS and the proportion of patients which obtained a MCID in SNOT−22 was performed. No statistically significant difference between groups (*p* = 0.11) could be demonstrated (Table 3).

## 4. Discussion

This is one of the earliest real-world studies on nasal symptoms in asthmatic patients treated with anti-IL-5-mAb [7]. Few anti-IL-5 drugs for CRSwNP treatment are currently authorized for commercialization. Regarding the anti-IL-5 inflammation route, only mepolizumab was approved by FDA for CRSwN treatment. Two other drugs were tested through a phase III randomized clinical trial (RCT): mepolizumab (SYNAPSE) [8] and benralizumab (OSTRO) [5], while reslizumab through a second phase RCT.

Despite all recent trials on mAb therapy excluding CRS without nasal polyps, about one third of these patients have eosinophilic inflammation [9]. In consequence, anti-IL-5 mAb therapy may have benefits controlling sinonasal symptoms also in patients without nasal polyps. This study, as well as the previous study carried out by Bajpai et al. [7], were performed in asthmatic patients with and without nasal polyps. Our results support this hypothesis.

Data on adverse effects related to these new mAb drugs raise the need for evaluating their safety. As eosinophils are known to have potential anti-parasitic and anti-viral effects, there are some concerns about the decrease in host defense during anti-IL-5 therapy. However, previous systematic reviews [10] and clinical trials about CRSwNP treatment with anti-IL-5 found no increase in the risk of infection [5,8,11]. In fact, the frequency of upper-airway infection in mAb treated patients was lower than that in patients who received a placebo. None of the previous RCTs in asthmatic patients have shown an increase in the infection rate in patients on mAb therapy [12,13]. Our findings are also in line with those of previous studies, whereby we did not observe an increase in the prevalence of respiratory tract infections.

In the present study, the change in SNOT−22 score after mAb treatment suggested a statistically significant clinical response to mAb. The two phase III RCTs on mepolizumab (SYNAPSE) and benralizumab (OSTRO) [5,8] had also applied the SNOT−22 questionnaire to assess nasal symptoms. The SNOT−22 baseline score in our study (57) was slightly lower than that reported in SYNAPSE (63.7) [8] and OSTRO (69.3) [5]. We also obtained a smaller change in SNOT−22 score before and after treatment with mAb than that in SYNAPSE [8] (20 vs. 29.4). Nevertheless, the difference in SNOT−22 score reported in our study was bigger than that found in the OSTRO study [5]. Interestingly, our results were similar to that reported in the other available anti-IL-5 real-life study performed by Bajpai et al. [7], were they reported 19.1 reduction in the SNOT−22. This divergence between studies could have several explanations. First, the studies vary in mAb treatment duration. In our study, the period of mAb treatment oscillates between 24 and 104 weeks, while it is 52 weeks in the SYNAPSE study [8], and 56 in the OSTRO study [5]. Second, not all anti-IL-5-mAb drugs might have the same influence on nasal symptoms. The patients in our study had received three types of anti-IL-5-mAb drugs. Third, carrying out the study in diverse populations might affect the obtained findings. We have assessed the effect of anti-IL-5-mAb in asthmatic patients with and without nasal polyps, Bajpai et al. [7] in asthmatic patients with CRS, while the SYNAPSE and OSTRO studies focused on CRSwNP patients exclusively.

Despite reslizumab, benralizumab and mepolizumab sharing the common final event of suppressing IL-5, their mechanism is different. Mepolizumab and reslizumab bind with to IL-5, thus preventing its interaction with the IL-5 receptor. However, benralizumab also promotes the eosinophil apoptosis induced by activation of the FcγRIIIa receptor of NK cells. This different mechanism of action could be related to different outcomes. However, despite our data suggests a slight better outcome for benralizumab compared to reslizumab and mepolizumab (Figure 2), the differences between groups were not statistically significant. This lack of significance could be attributed to the scarce sample size, which will be improved in the following years.

Not all patients under mAb treatment show improved nasal symptoms [11]. This suggests that each type of CRSwNP might require a specific treatment according to the pathophysiology of the disease. Accordingly, medication schemes adapted to disease phenotype are needed. Up to date, all mAb prescriptions have similar indications [2,14], despite available evidence suggesting that it may not be the best approach.

Accordingly, in this study, we explored various potential predictors of a good clinical response in order to better select the candidates for anti-IL-5 mAb treatment. We observed that blood eosinophils, rather than serum total IgE, is the best predictor of improvement in asthma and nasal symptoms assessed by ACT and SNOT−22 questionnaires.

Current guidelines recommend mAb treatment in case of increased serum total IgE and/or tissue or blood eosinophil levels [2,14], without referring to any specific indication on the different anti-IL-5 mAb drugs. In this study, eosinophils seem to be a better predictor than serum total IgE in the nasal clinical response to mAb. Nonetheless, this finding should be considered preliminary until its replication in larger studies and different populations.

Opposed to our results, RCTs with mepolizumab [8,15,16] or reslizumab [11] have shown no association between baseline blood eosinophil count and achieving better clinical improvement. Interestingly, the OSTRO phase III RCT study on benralizumab suggested that response to treatment might be influenced by the baseline blood eosinophil count, despite that the undertaken interaction analysis did not reach nominal significance [5].

We also reported that patients’ weight before mAb is significantly related to the change in the SNOT−22 score before and after mAb treatment. The same tendency, despite not reaching statistical significance, was observed for the change in ACT score. It means that, the higher the patients’ weight before mAb treatment, the lesser the change in the SNOT−22 score. However, we did not observe a statistically significant association between BMI and the change in ACT or SNOT−22 scores before and after mAb treatment. This finding raises the following two hypotheses. First, obesity may worsen the prognosis of anti-IL-5-mAb therapy. There is no published evidence on BMI influence in CRSwNP patients, but the association of obesity with poor response to mAb therapy in asthmatic patients is well established [17,18]. Second, the observed effect of weight in response to mAb treatment may be related to mAb dosage. In this study, mepolizumab was administered subcutaneously, while in other studies, it was given intravenously [15]. When given subcutaneously, mAb dosage is always constant; however, the dosage can be weight-adjusted when the drug is administered intravenously. To the best of our knowledge, only the OSTRO study [5] was carried out in CRSwNP patients and showed evidence of a differential effect of mAb based on BMI, but the interaction tests did not reach nominal significance [5].

In asthmatic patients, a real-world study reported a decreased response to mepolizumab with the increase in BMI [19]. The authors of that study attribute this finding to the fact that mepolizumab is always administered at the same dosage, 100 mg, which could be insufficient for treating asthmatic obese patients [20]. Consequently, future research is required to investigate whether mAb dosage should be weight-adjusted for better control of asthma symptoms. It is noteworthy that our population encompassed more obese patients than that of the previous RCTs, as we had an average BMI of 28.54, while previous studies reported an average BMI between 24 and 26 [5,15]. BMI average was not reported in all studies [11,16].

We did not identify any significant association between the presence of N-ERD diagnosis and SNOT−22 change after the mAb therapy. The only available evidence (none in anti-IL-5-mAb) does not report worse results in patients suffering from N-ERD [21]

We reported a minute and non-statistically significant difference in the SNOT−22 change before and after mAb treatment regarding the presence of previous ESS (25.33 vs. 14.21, *p* = 0.11). We even found a higher proportion of patients reaching MCID for SNOT−22 in cohorts of patients with small polyps (I and II), in comparison with big polyps (III). There is an extensive debate on whether there is a role for mAb in patients without previous ESS. EPOS guidelines indicate mAb treatment for patients with previous ESS exclusively [2]; however, EUFOREA guidelines also recommend mAb for patients with other clinical characteristics [14]. Evidence from available literature demonstrated a relatively modest reduction in polyp size, suggesting that a considerable fraction of patients might still need ESS despite being treated with mAb [10,11]. Future studies are required to address this question.

Finally, we observed that the size of the polyps was significantly related to the blood eosinophil baseline count. We also found that polyp size was neither related to serum total IgE baseline levels, nor to the change in ACT or SNOT−22 (globally and MCID) scores before and after mAb treatment. Our findings on the absence of association between the size of the polyps and the pretreatment SNOT−22 score contradicts that of the SYNAPSE study [8], which reported a correlation between NPS and the SNOT−22 and the visual analogue scale (VAS) scores. Most RCTs suggest mAb treatment in patients with polyps scores III and IV [11,15,16], but our real-world data revealed poorer response in these patients which could be due to their nasal symptoms.

### Limitations

The main limitation of our study is the sample size; however, this was expected as mAb therapy for CRSwNP is a novel approach, and most heath care centers have few patients under this treatment. The limited sample size led to the absence of statistically significant associations; hence, future larger studies are needed to confirm our findings. Second, this is an observational study. It means that it is an uncontrolled sample, which may introduce some observational bias.

The second limitation is that not all the participants have CRSwNP. The objective of this study was not assessing the effect of mAb therapy on nasal polyps, which has been widely studied. The objective was only assessing nasal symptoms instead, independently of the diagnosis of nasal illnesses.

The third limitation is that we have not included the whole spectrum of polyp size, as no patients with NPS IV were included in this study. As this data came from daily practice, this may reflect the fact that NPS IV may have been deemed candidates of surgery before monoclonal antibody therapy.

## 5. Conclusions

This real-life study has reported a significant clinical response based on SNOT−22 score after anti-IL-5 mAb treatment. This study demonstrated that blood eosinophil count, rather than serum total IgE levels, is the best predictor of nasal symptom improvement, this assessed through the SNOT−22 and the asthmatic symptoms through ACT. Two research questions to guide future studies arose from our findings: if anti-IL-5 mAb for CRSwNP should be dose-adjusted by weight, and if the indication of anti-IL-5 mAb for CRSwNP should always require a high baseline blood eosinophil.

## Figures and Tables

**Figure 1 jcm-11-07056-f001:**
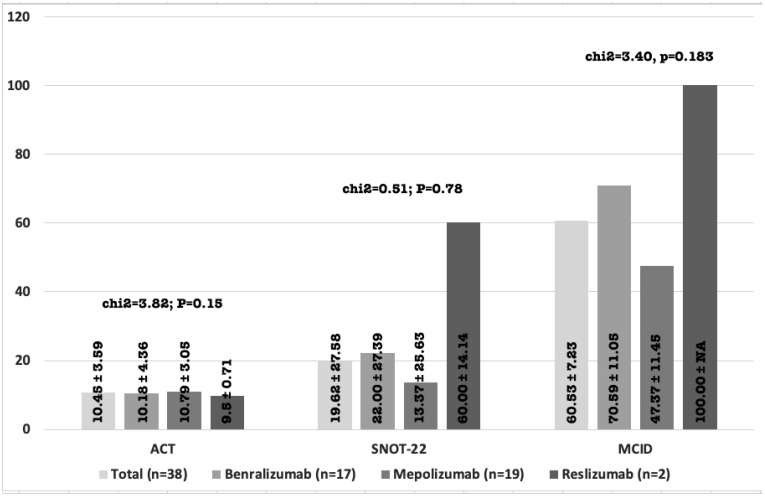
Clinical response to anti-IL-5 mAb therapy. Data in percentage of change. MCID (minimal clinically important difference). NA (Not applicable). Statistical analysis was performed comparing the difference between the different anti-IL-5 mAb.

**Figure 2 jcm-11-07056-f002:**
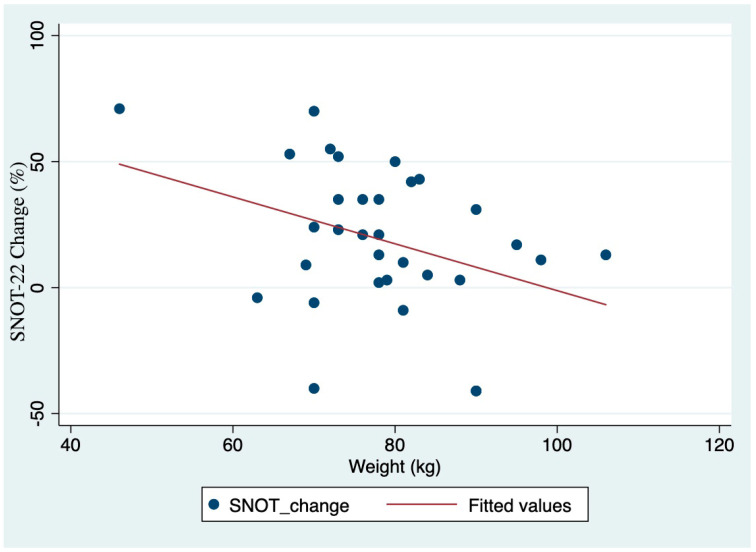
Correlation between the change in SNOT−22 questionnaire and body weight.

**Table 1 jcm-11-07056-t001:** Description of the sample. Bold and asterisk if statistically significant.

Comparison between Groups	Age (Mean ± SD) (Range)	Male (n, %)/Female (n, %)	Weight/BMI (Mean ± SD)	FESS (n, %)	Serum Total IgE (Mean ± SD)	Blood Eosinophils (Mean ± SD)
Total (n = 38)	56.66 ± 9.00 (33-79)	16 (42.11%)/22 (57.89%)	78.76 ± 11.33/ 28.54 ± 4.68	19 (50%)	443.53 ± 646.32	655.92 ± 478.05
Benralizumab (n = 17)	54.53 ± 10.82 (33-79)	7 (41.18%)/10 (58.82%)	76 ± 13.85/ 28.11 ± 6.24	6 (35.29%)	746.25 ± 891.98	578.69 ± 343.66
Mepolizumab (n = 19)	58.26 ± 7.16 (46-68)	8 (42.11%)/11 (57.89%)	81.81 ± 8.49/ 28.89 ± 2.08	11 (57.89%)	253.6 ± 306.03	735.26 ± 584.63
Reslizumab (n = 2)	59.5 ± 7.78 (54–65)	1 (50%)/1 (50%)	75 ± 7.07/ 28.95 ± 9.34	2 (100%)	128.75 ± 85.21	520 ± 226.27
Statistical analysis	chi^2^ = 2.45 *p* = 0.294	chi^2^ = 0.04; *p* = 0.844	chi^2^ = 3.12; *p* = 0.210	chi^2^ = 0.82; *p* = 0.364	**chi^2^ = 16.25;** ***p* < 0.001 ***	chi^2^ = 4.26; *p* = 0.119

**Table 2 jcm-11-07056-t002:** Predictors of clinical response.

Pre-Treatment Variables	ACT Change	SNOT−22 Change
**IL-5 mAb dose**	Rho = −0.02 *p* = 0.483	Rho = 0.12 *p* = 0.482
**Weight**	Rho = 0.29 *p* = 0.103	Rho = −0.36 *p* = 0.041
**Height**	Rho = 0.13 *p* = 0.480	Rho = −0.13 *p* = 0.459
**Age**	Rho = 0.07 *p* = 0.671	Rho = −0.04 *p* = 0.826
**BMI**	Rho = 0.18 *p* = 0.307	Rho = −0.23 *p* = 0.207
**Blood eosinophil**	Rho = −0.32 *p* = 0.056	Rho = 0.30 *p* = 0.078
**Total serum IgE**	Rho = −0.20 *p* = 0.246	Rho = 0.02 *p* = 0.927

**Table 3 jcm-11-07056-t003:** Relationship between the size of the polyps with different variables. MCID (minimal clinically important difference). NA (not applicable) Bold and asterisk if the difference is statistically significant (*p* < 0.05).

Polyp Size		None (n = 13)	I (n = 3)	II (n = 11)	III (n = 11)	Statistical Analysis
**Serum total IgE**		424.49 ± 741.64	300.5 ± 89.80	326.02 ± 602.18	608.31 ± 686.14	chi^2^ = 3.08; *p* = 0.379
**Eosinophils**		520 ± 317.22	693.33 ± 457.20	728.9 ± 349.74	758.01 ± 751.99	chi^2^ = 8.86; *p* = 0.03 *
**SNOT−22 change**		13.54 ± 23.79	28 ± 19.31	32.45 ± 29.44	13.11 ± 30.56	chi^2^ = 1.05; *p* = 0.789
**ACT change**		−10.23 ± 2.98	−11 ± 1.73	−11.18 ± 2.82	−9.2 ± 5.07	chi^2^ = 5.41; *p* = 0.144
**SNOT−22 pre**		49.08 ± 32.04	77.67 ± 12.06	63.27 ± 17.64	48.8 ± 24.42	chi^2^ = 4.69; *p* = 0.196
**SNOT−22 MCID**		53.85 ± 13.83	66.67 ± 27.22	81.82 ± 11.63	50.00 ± 15.81	chi^2^ = 2.84, *p* = 0.416
**Smoking**	**No**	8 (61.54)	1 (33.33)	7 (63.64)	6 (54.55)	chi^2^ = 2.84; *p* = 0.829
	**Former**	4 (30.77)	1 (33.33)	2 (18.18)	4 (36.36)	
	**Yes**	1 (7.69)	0 (0.00)	0 (0.00)	0 (0.00)	
	**Missing**	0 (0.00)	1 (33.33)	2 (18.18)	1 (9.09)	
**Alcohol**		1 (7.69)	1 (33.33)	2 (18.18)	0 (0.00)	chi^2^ = 3.58; *p* = 0.310
**Weight**		73.92 ± 5.66	98 ± NA	77.2 ± 7.96	84.56 ± 16.59	chi^2^ = 11.06; *p* = 0.004 *
**Asthma**		13 (100%)	3 (100%)	11 (100%)	11 (100%)	NA
**N-ERD**		2 (15.38%)	1 (33.33%)	1 (9.09%)	4 (36.36%)	chi^2^ = 2.01; *p* = 0.571
**Age**		58.38 ± 8.20	57.67 ± 13.87	54.27 ± 10.53	57.1 ± 8.03	chi^2^ = 14.54; *p* = 0.069

## Data Availability

Not applicable.

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
