# Peer review of "Nasal Symptoms in Asthmatic Patients under Treatment with Anti-IL-5 Monoclonal Antibodies. A Real-Life Cohort Study"

_jcm, 2022, doi:10.3390/jcm11237056_

Round 1
Reviewer 1 Report
The authors have presented a study about "Nasal symptoms in asthmatic patients under treatment with anti-IL-5 monoclonal antibodies. A real-life cohort study". The manuscript is really interesting, with a real-life cohort analyzed instead of randomized controlled trial. There are some suggestions that could be made to increase the overall quality.
1. Title: Please type IL-5 instead of IL5. This should be applied to the whole manuscript.
2. In the sample section (Lines 69 - 87): Were only CRSwNP patients combined with asthma were included in this study? How do the authors define CRSwNP? In addition, if CRSsNP patients were included in this study, the authors should explain the reason in detail. Although the statement that this study was performed among uncontrolled samples was described in the limitation section of the Discussion, please describe inclusion criteria in detail in the first part of the Sample section of Materials and Methods.
3. In the mAb therapy section (Lines 111 - 119): The authors only described the laboratory criteria for mAb in patients with asthma. Readers might be curious about the clinical indication of mAb for asthma. Please, provide the clinical criteria if it exists in the local guidelines, or sample cases.
4. Table 1 (Line 140): Data for turbinate surgery and septoplasty is not important in this study. Eliminating or at least moving to the data to the supplementary materials, would be more appropriate. In addition, please add “Serum total” IgE and “Blood” Eosinophils. (This should also be applied to the text.)
5. In the predictors of clinical response (Lines 172 - 189) and Nasal polyp score (Lines 191 - 201) sections: The authors described detailed scores in the text (e.g. SNOT-22 change (rho-0.30; p=0.07)) and a relationship between NPS and the SNOT-22 change (chi2=4.69, p=0.20). Unfortunately, it is impossible to find the detailed data in the table, the figure or in supplementary materials section. Please clarify this or state, “data not shown”.
6. Please revise units, the names of variables, and typos: SNOT-22 change (from SNOT 22 change), SNOT-22 pre (from SNOT22 pre), SNOT-22 MCID (from SNOT22 MCID), chi2 (from chi2), Asthma (from Asma: typo in Table 3) and CRSwNP (from CRswNP).
7. Table 2: This table contains the key findings of this study regarding the clinical response to anti-IL-5 mAb therapy. However, data in the table is counterintuitive. Rather than a table, please consider using a figure for better readability.
8. Table 3: The authors stated a total sample of 38 patients were included. However, the sum of n was 43. In addition, were there no NPS IV patients? It would be interesting to know the efficacy of IL-5 mAb in patients with huge polyps. Also, the number of smoking variable in NPS I (n=11) were inaccurate. The sum of “No smoking (n=1)”, “Former” (n=1), and “Yes smoking” (n=0) is just 2, not 11. These numbers in NPS II were also inaccuate. Please, clarify these numbers.
9. Figure 1: Please provide the unit of Y-axis. (probably SNOT-22 change)
10. In discussion section: Among three IL-5 mAb drugs, benralizumab (anti-IL-5Ra mAb) uses a different immunologic route compared to mepolizumab and reslizumab (anti-IL-5). The authors should discuss the meaning of this different immunologic pathway and any related data from the present study.
11. In discussion section (Lines 223 - 232): The authors discussed the safety issues regarding IL-5 mAb drugs. The authors mentioned the prevalence of URI in the discussion section. However, data related to the prevalence of URI was not provided in this present study. If additional safety data could not be provided, the authors cannot emphasize the safety of mAb.
12. In discussion (Lines 256 - 275): The authors described in detail potential predictors in long paragraphs. However, the knowledge regarding predictors is not yet conclusive even though the blood eosinophil count is considered a good predictor of efficacy. Hence, shortening the descriptions of predictors would be better in the discussion section of this present study.
13. Obesity and BMI (Lines 276 - 292): The authors explained the reason why BMI was not associated with SNOT-22 change. The authors mentioned that the association of obesity with poor response to mAb therapy in asthmatic patients was well established in literature whereas there are no published articles regarding BMI influence in CRSwNP. However, this is not an appropriate or direct reason for that.
14. In discussion section (Line 300): Please verify the number “265,15”. It probably should be 26 5,15 (the reference number being 5, 15). Please use MDPI reference style using brackets (i.e. [5,15]) in whole manuscript.
Author Response
REVIEWER 1
The authors have presented a study about "Nasal symptoms in asthmatic patients under treatment with anti-IL-5 monoclonal antibodies. A real-life cohort study". The manuscript is really interesting, with a real-life cohort analyzed instead of randomized controlled trial. There are some suggestions that could be made to increase the overall quality.
We sincerely appreciate the effort reviewing our manuscript. We find the corrections apropiate, and we feel that the manuscript has now increased its quality. Most of the authors are also reviewers, and we know the time it takes to review a manuscript the way it has been done.
- Title: Please type IL-5 instead of IL5. This should be applied to the whole manuscript.
- Done
- In the sample section (Lines 69 - 87): Were only CRSwNP patients combined with asthma were included in this study? How do the authors define CRSwNP? In addition, if CRSsNP patients were included in this study, the authors should explain the reason in detail. Although the statement that this study was performed among uncontrolled samples was described in the limitation section of the Discussion, please describe inclusion criteria in detail in the first part of the Sample section of Materials and Methods.
- Not all the asthmatic patients had CRSwNP. It has been explained in the methods (page 2, line 79) and the limitations section (page 9, line 359-362)
- In the mAb therapy section (Lines 111 - 119): The authors only described the laboratory criteria for mAb in patients with asthma. Readers might be curious about the clinical indication of mAb for asthma. Please, provide the clinical criteria if it exists in the local guidelines, or sample cases.
- Line 117-119
- Table 1 (Line 140): Data for turbinate surgery and septoplasty is not important in this study. Eliminating or at least moving to the data to the supplementary materials, would be more appropriate. In addition, please add “Serum total” IgE and “Blood” Eosinophils. (This should also be applied to the text.)
- We agree with the reviewer, it has been modified in the table (table 1, line 157).
- “Serum total” and “blood” were included throughout the text.
- In the predictors of clinical response (Lines 172 - 189) and Nasal polyp score (Lines 191 - 201) sections: The authors described detailed scores in the text (e.g. SNOT-22 change (rho-0.30; p=0.07)) and a relationship between NPS and the SNOT-22 change (chi2=4.69, p=0.20). Unfortunately, it is impossible to find the detailed data in the table, the figure or in supplementary materials section. Please clarify this or state, “data not shown”.
- It has been included (table 2, line 201)
- Please revise units, the names of variables, and typos: SNOT-22 change (from SNOT 22 change), SNOT-22 pre (from SNOT22 pre), SNOT-22 MCID (from SNOT22 MCID), chi2(from chi2), Asthma (from Asma: typo in Table 3) and CRSwNP (from CRswNP).
- We appreciate the observation; the excellence is in the details. It has been modified throughout the document.
- Table 2: This table contains the key findings of this study regarding the clinical response to anti-IL-5 mAb therapy. However, data in the table is counterintuitive. Rather than a table, please consider using a figure for better readability.
- It has been transformed into a figure (figure 1).
- Table 3: The authors stated a total sample of 38 patients were included. However, the sum of n was 43. In addition, were there no NPS IV patients? It would be interesting to know the efficacy of IL-5 mAb in patients with huge polyps. Also, the number of smoking variable in NPS I (n=11) were inaccurate. The sum of “No smoking (n=1)”, “Former” (n=1), and “Yes smoking” (n=0) is just 2, not 11. These numbers in NPS II were also inaccuate. Please, clarify these numbers.
- We truly appreciate this correction. There was some mistakes in the table. Now it is correct. Regarding the data of smoking habit, as this data came from daily practice, some of them were missing data (now included in the table 3).
- In this cohort we have not NPS IV. This has been added to the limitation section (line 365-368).
- Figure 1: Please provide the unit of Y-axis. (probably SNOT-22 change)
- Yes, it is correct. It has been added to the figure (now figure 2).
- In discussion section: Among three IL-5 mAb drugs, benralizumab (anti-IL-5Ra mAb) uses a different immunologic route compared to mepolizumab and reslizumab (anti-IL-5). The authors should discuss the meaning of this different immunologic pathway and any related data from the present study.
- It has been included in the discussion section (267-275)
- In discussion section (Lines 223 - 232): The authors discussed the safety issues regarding IL-5 mAb drugs. The authors mentioned the prevalence of URI in the discussion section. However, data related to the prevalence of URI was not provided in this present study. If additional safety data could not be provided, the authors cannot emphasize the safety of mAb.
- Yes, URI has been recorded in this study (methods line 85-86), however, it has not been previously reported in the results section. Now it has been added (line 174-176).
- In discussion (Lines 256 - 275): The authors described in detail potential predictors in long paragraphs. However, the knowledge regarding predictors is not yet conclusive even though the blood eosinophil count is considered a good predictor of efficacy. Hence, shortening the descriptions of predictors would be better in the discussion section of this present study.
- In order to shorten this section, we have eliminated the paragraph related to asthma studies, as our study was focused on nasal symptoms (line 297-303).
- Obesity and BMI (Lines 276 - 292): The authors explained the reason why BMI was not associated with SNOT-22 change. The authors mentioned that the association of obesity with poor response to mAb therapy in asthmatic patients was well established in literature whereas there are no published articles regarding BMI influence in CRSwNP. However, this is not an appropriate or direct reason for that.
- Sorry, but we don’t understand this correction. As there is no evidence in this regard, we cannot discuss or compare our data to other’s.
- In discussion section (Line 300): Please verify the number “265,15”. It probably should be 26 5,15 (the reference number being 5, 15). Please use MDPI reference style using brackets (i.e. [5,15]) in whole manuscript.
- The references has been modified following the Zotero citation style for Multidisciplinary Digital Publishing Istitute.
Reviewer 2 Report
Thank you for the opportunity to review this study on real world practice of biologics.
As there is still not much information on the benefits of using biologics in real world practice, this study addresses this gap.
I have the following comments:
1. I am concerned that the patient's population is not uniform and properly standardized. Patients with or without surgery were mixed together and assumed to have similar characteristics. I would like to suggest the authors to do a subgroup analysis on the outcomes of these 2 groups of patients ie with or without surgery.
2. I find it very hard to read Table 1 to 3, as they are constructed poorly and very confusing. Please improve on the presentation of the data in all tables to make them better understood. Moreover, why is there a need to have 2 statistical analyses in Table 2?
3. There is a disconnect between the objectives of the study and conclusions. The authors aimed to describe the nasal effects of adult asthmatic cohort 65 treated with anti-IL-5 mAb for CRswNP but the conclusions were "This study demonstrated that eosinophil count, rather than IgE, is the best pre-335 dictor of asthma symptom improvement assessed through the SNOT-22 and the ACT 336 questionnaires". This sentence also needs to be rephrased "A significant clinical response based on SNOT-22 score evolution after anti-IL-5 mAb 334 treatment" as it is not clear.
Author Response
REVIEWER 2
Thank you for the opportunity to review this study on real world practice of biologics.
We appreciate the time and effort spent reviewing our manuscript.
As there is still not much information on the benefits of using biologics in real world practice, this study addresses this gap.
I have the following comments:
- I am concerned that the patient's population is not uniform and properly standardized. Patients with or without surgery were mixed together and assumed to have similar characteristics. I would like to suggest the authors to do a subgroup analysis on the outcomes of these 2 groups of patients ie with or without surgery.
- This is a valid concern, which we also share. This is the main problem in non-controlled studies. In our first analysis we have already tried the analysis the reviewer suggest, but as the sample size is too small, we don’t find any significant difference between groups, and we lose some of the associations that we find in a larger group. In fact, it has been already included in the discussion (line 333-335 “We reported a minute and non-statistically significant difference in the SNOT-22 change before-and after-mAb treatment regarding the presence of previous ESS (25.33 versus 14.21, p= 0.11).”
- I find it very hard to read Table 1 to 3, as they are constructed poorly and very confusing. Please improve on the presentation of the data in all tables to make them better understood. Moreover, why is there a need to have 2 statistical analyses in Table 2?
- Table 1 was shortened
- In table 2 (now removed), the rows were exploring the differences in ACT and SNOT-22 for each monoclonal antibody. The columns, were exploring the differences between the three different monoclonal antibodies. We understand that is may be confusing to read. As it has also been suggested by other reviewer, we decided to transform this table into a figure (now figure 1).
- Table 3 had some mistakes that have now been corrected.
- There is a disconnect between the objectives of the study and conclusions. The authors aimed to describe the nasal effects of adult asthmatic cohort treated with anti-IL-5 mAb for CRswNP but the conclusions were "This study demonstrated that eosinophil count, rather than IgE, is the best predictor of asthma symptom improvement assessed through the SNOT-22 and the ACT 336 questionnaires". This sentence also needs to be rephrased "A significant clinical response based on SNOT-22 score evolution after anti-IL-5 mAb treatment" as it is not clear.
- We agree with the reviewer. It has been modified (line 370-376).
Round 2
Reviewer 1 Report
It is clear that the authors have made an effort to respond to each the reviewer’s suggestions, which is greatly appreciated. However, there are still several points that may require revision or clarification.
This real-life cohort study regarding anti IL-5 monoclonal antibodies is interesting and the manuscript has definitely improved.
Here are some minor suggestions.
1. The authors stated “Not having chronic rhinosinusitis was not an exclusion criteria”. This double negative expression may be a bit confusion. Please, revise this sentence in a more easily comprehensible way.
2. Table 1: Please add a “/” between scores in Weight/ BMI. (i.e. 78.76 ± 11.33 / 28.54 ± 4.68)
3. In the manuscript, the authors described SNOT-22 before ACT whereas they presented ACT first in the table and the figure. It would be best to maintain consistency in the order. It seems that presenting ACT first and SNOT-22 second would be most appropriate in the text, so this order should be applied throughout the whole manuscript.
4. Table 3: please add a “-“ in “SNOT-22” (instead of “SNOT 22”). This should be applied throughout the whole text.
5. Table 3: please harmonize and reformat the font size. In particular, “chi2=3.08” and “p=0.379” are written in a much bigger font size than other numbers in the Statistical analysis of Serum total IgE.
6. Figure 2: It seems that the title of the Y-axis has not yet been added, “SNOT-22 change”, like “Weight (Kg)” on the X-axis.
Author Response
It is clear that the authors have made an effort to respond to each the reviewer’s suggestions, which is greatly appreciated. However, there are still several points that may require revision or clarification.
Thank you again for your comments.
This real-life cohort study regarding anti IL-5 monoclonal antibodies is interesting and the manuscript has definitely improved.
Here are some minor suggestions.
- The authors stated “Not having chronic rhinosinusitis was not an exclusion criteria”. This double negative expression may be a bit confusion. Please, revise this sentence in a more easily comprehensible way.
- It has been modified (line 80-81)
- Table 1: Please add a “/” between scores in Weight/ BMI. (i.e. 78.76 ± 33 /28.54 ± 4.68)
- It has been added (table 1, line 161)
- In the manuscript, the authors described SNOT-22 before ACT whereas they presented ACT first in the table and the figure. It would be best to maintain consistency in the order. It seems that presenting ACT first and SNOT-22 second would be most appropriate in the text, so this order should be applied throughout the whole manuscript.
- The order has been modified throughout the whole manuscript (abstract line 27-39; line 170-174; 188-190; 192-193; 195-197; 288; 307; 344;
- Table 3: please add a “-“ in “SNOT-22” (instead of “SNOT 22”). This should be applied throughout the whole text.
- Added (table 3, line 226).
- Table 3: please harmonize and reformat the font size. In particular, “chi2=3.08” and “p=0.379” are written in a much bigger font size than other numbers in the Statistical analysis of Serum total IgE.
- It has been corrected (table 3, line 226)
- Figure 2: It seems that the title of the Y-axis has not yet been added, “SNOT-22 change”, like “Weight (Kg)” on the X-axis.
- Now is correct
Reviewer 2 Report
Authors have addressed concerns raised. Thank you.
Author Response
Thank you again for reviewing our manuscript.
As it has been pointed out that "English language and style are fine/minor spell check required" the manuscript has been checked again by a profesional english translator.